# Enhancing Structural Performance of Short Fiber Reinforced Objects through Customized Tool-Path

**Jaeyoon Kim and Bruce S. Kang** * 

Department of Mechanical and Aerospace Engineering, West Virginia University, Morgantown, WV 26505, USA; jekim@mix.wvu.edu
* Correspondence: bruce.kang@mail.wvu.edu; Tel.: +1-304-293-3232

**Abstract:** Fused deposition modeling (FDM) is one of the most common additive manufacturing (AM) technologies for thermoplastic materials. With the development of carbon fiber-reinforced polymer (CFRP) filament for FDM, AM parts with improved strength and functionality can be realized. CFRP is anisotropic material and its mechanical properties have been well studied, however, AM printing strategy for CFRP parts has not been developed. This paper proposes a systematic optimization of the FDM 3D printing process for CFRP. Starting with standard coupon specimen tests to obtain mechanical properties of CFRP, finite element analyses (FEA) were conducted to find principal directions of the AM part and utilized to determine fiber orientations. A specific tool-path algorithm has been developed to distribute fibers with the desired orientations. To predict/assess the mechanical behavior of the AM part, the 3D printing process was simulated to obtain the anisotropic mechanical behavior induced by the customized tool-path printing. Bolt hole plate and spur gear were selected as case studies. FE simulations and associated experiments were conducted to assess their performance. CFRP parts printed by the optimized tool-path shows about 8% higher stiffness than those printed at regular infill patterns. In summary, assisted by FEA, a customized 3D printing tool-path for CFRP has been developed with case studies to verify the proposed AM design optimization methodology for FDM.

**Keywords:** Fused Deposition Modeling (FDM); tool-path; FEA; CFRP

## 1. Introduction

Additive manufacturing (AM) technologies have been rapidly advancing and widening its applicability to complex geometries and range of material choice [1–6]. Fused deposition modeling (FDM) is one of the widely used in AM technologies for the thermoplastic material. In contrast to traditional subtractive manufacturing, FDM parts are built by adding materials layer by layer. This layered nature of FDM causes some defects of the printed objects, such as staircase effect, coarse surface, and anisotropic mechanical properties [3,4]. To address these challenges, improvement of the quality of FDM parts has been an active research area. For example, printing process techniques, such as heat treatment [7], machining [8], and chemical treatment [9–12], were investigated to have a better surface quality of the parts. Printing parameters, including the effect of raster angle [13] and building direction [14], were optimized to obtain better bonding strength. However, there is a lack of design strategy to address the anisotropic characteristics of FDM parts. Moreover, the development of carbon fiber-reinforced polymer (CFRP) for FDM requires the optimization of material anisotropy for its best performance. For CFRP, extensive research has been carried out to investigate the anisotropic mechanical properties of CFRP, including ABS, PLA, and nylon. Generally, the anisotropic structural property of FDM parts is highly dependent on the building direction. Short carbon fiber provides 3D printing flexibility to improve the mechanical properties of parts than those printed by continuous

fiber-reinforced filaments [15]. The effect of build orientation has been studied to enhance structural performance [16–18]. For FDM 3D printing, commercial slicer programs provide several limited infill patterns to choose a building direction. Once a pattern is selected, it is not allowed to edit its route. This is because the tool-path for FDM was originally developed to control the movement of the CNC machine cutter. When the tool-path algorithm was developed for FDM, it was for printing process improvement, structural strength enhancement of final products was not recognized. In this research, a novel design methodology for short carbon fiber composite built by FDM 3D printing is proposed. It starts with FEA stress analysis to determine principal stress directions of the FDM part with which customized tool-paths are developed. Structural performances of FDM parts built by the proposed tool-path method are verified by both FEA analyses and experiments. Tensile tests and related microstructural analyses using scanning electron microscope (SEM) for CFRP-Nylon were performed to investigate mechanical properties and fiber orientations. The framework for this methodology is shown in Figure 1. The primary contributions of this research are as follows:

(1) A novel design methodology for FDM parts assisted by finite element analysis;
(2) A customized tool-path algorithm for FDM that maximizes the effect of fiber reinforcement under the given loading and boundary conditions.

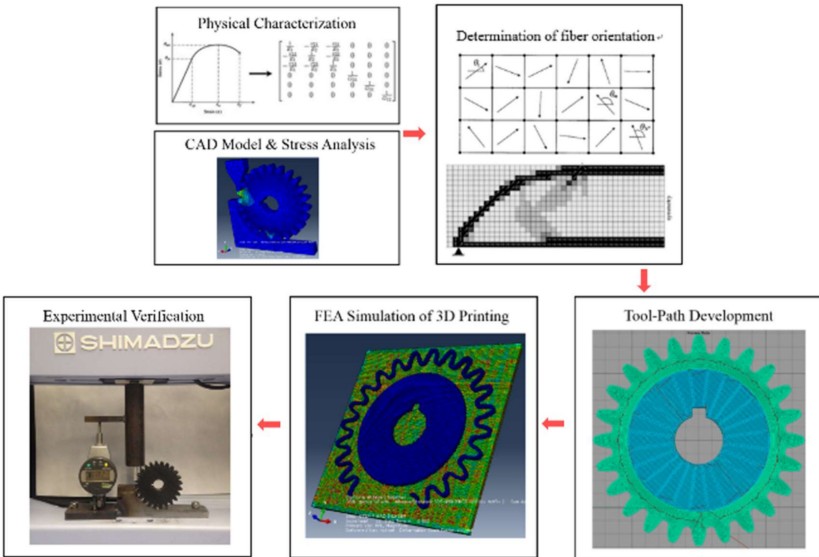

**Figure 1.** The workflow of the design approach.

## 2. Background

### 2.1. Anisotropy of FDM Parts

Many experimental studies have shown that FDM printed parts exhibit anisotropic mechanical properties. Ahn et al. investigated the anisotropic mechanical properties of ABS parts built by FDM [17]. The mechanical properties of PEEK and PC fabricated by FDM were studied by El-Gizawy et al. [19] and Hill and Haghi [20], respectively. They proved that FDM induces a significant structural anisotropy for thermoplastic materials.

There have been several computational methods to address this issue. Hildebrand et al. proposed a computational design methodology to apply different build orientations to each section of an FDM part based on its stress map [20]. Zhou et al. developed a worst-case analysis method assuming the material is orthotropic to find the structurally weak sections [21]. Farbman et al. reported the effect of infill pattern on the mechanical properties of FDM parts by FEA simulation [22].

Besides, build orientation selection methods to control the anisotropy has been proposed. Domingo-Espin et al. investigated the effect of building direction by FEA modeling of FDM parts with

orthotropic materials [23]. Richard and Crawford developed a build orientation selection algorithm using the Tsai–Wu failure criterion [24]. However, their works did not try to find the build orientation to minimize internal stress. Similarly, Umetani and Schmidt proposed a method to find the best build orientation for FDM parts [25]. This method accounted for structural anisotropy to stacking layers of extrusion, assuming those are laminate composites. They found that different level of structural performances was captured depending on build direction.

## 2.2. Carbon Fiber Reinforced Polymer (CFRP)

In 3D printing, various fiber-reinforced polymers have been studied to improve mechanical properties using Jute fibers [21], metal [22], glass fibers [23], vapor-grown carbon fiber [24], and continuous fibers [25]. For FDM 3D printing, short fiber reinforced filaments are most commonly used in manufacturing high strength AM part. It has been reported that short carbon fibers reinforced thermoplastic filaments significantly improve the strength of AM parts. These filaments are now commercially available from several manufacturers, such as CarbonX$^{TM}$, Matterhackers$^{TM}$, and ColorFabb$^{TM}$. Figure 2 illustrates a printing extrusion of a short fiber reinforced filament.

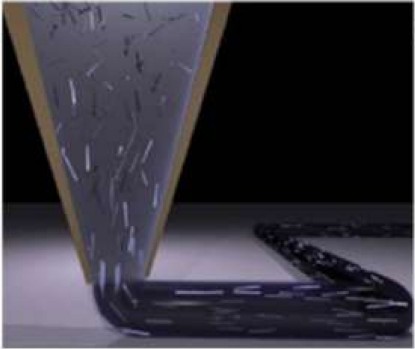

**Figure 2.** Short fiber alignment during the extrusion process [26].

## 2.3. Tool Path for FDM

It has been demonstrated that tool-path or build orientation affects the quality and strength of objects. Concerted efforts have been made to optimize tool-path planning for various applications [27,28]. Currently, contour-parallel and direction-parallel methods are the most widely used for FDM in practice. The contour-parallel tool-path is to move the extruder parallel to the boundaries of cross-sections [29]. Fabrication is precise, but it is computationally expensive. By contrast, the direction-parallel path is to move the extruder in zig-zag motion along a fixed direction within the boundary in the interior region. This method is simple and fast. The algorithm using both methods has been developed and is widely adopted in major slicer programs, such as Simplify3D [30]. Figure 3 illustrates the difference between the contour-parallel and directional-parallel tool-path.

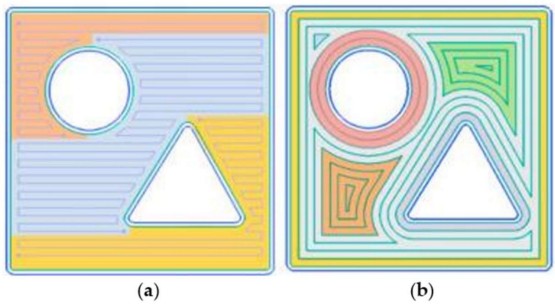

(a)          (b)

**Figure 3.** Comparison of different tool-path generation strategies; (**a**) direction parallel path, (**b**) contour parallel path, [31].

## 3. Material Characterization

### 3.1. Mechanical Property Measurement

As various materials for FDM have been developed, associated test methods have been reported. Nevertheless, American Society for Testing and Materials (ASTM) test protocols for additive manufactured objects officially has not been developed [32,33]. In this study, CFRP is treated as a laminated plastic polymer. CFRP-Nylon (20% fiber volume fraction) coupon specimens with printing orientation of 0°, 45°, and 90° were fabricated, and relevant mechanical properties were measured. As shown, direction 1 (red color) is the FDM line of deposition, and direction 2 is perpendicular to this line of deposition. From the printed specimens oriented at 0°, tensile modulus in the deposition direction $E_1$, Poisson ratio $\nu_{12}$ and tensile strength $S_1$ were determined. From the specimens oriented at 90°, tensile modulus perpendicular to the deposition direction $E_2$, Poisson ratio $\nu_{21}$ and tensile strength $S_2$ were determined. From the specimens oriented at ±45°, shear modulus at the 1–2 plane $G_{12}$ and shear strength $S_{12}$ were determined. Three specimens per sample were tested for each one of the three orientation cases, and each printing material, totalizing 9 test runs.

For the determination of stiffness and strength properties at material directions 1 and 2, the ASTM D638 test standard was applied. As shown in Figure 4, I-type (dog-bone) specimen was used, with length and width of 165 mm and 19 mm, respectively. Letter-size sheets (11' × 8.5") were printed by the designated printing orientations and cut by a CNC machine to obtain specimens. The nominal specimen thickness was 3.3 mm, with 11 printed layers. ASTM D3518 test method was conducted to obtain the 1–2 plane shear stiffness and strength properties. Rectangular specimens (25 mm × 200 mm) were printed. The nominal thickness is 4.8 mm with 16 printed layers, and the stacking sequence was [±45°]4s (symmetric).

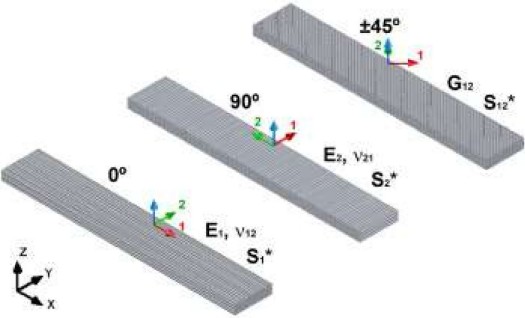

**Figure 4.** Illustration of printing orientations (0°, 90°, and ±45°) [34].

Ultimaker 2+ was used to produce test specimens. It was one of the popular 3D printers at the time of study, allowing users to upgrade it to apply CFRP filaments. The printing envelope of 215 mm × 210 mm × 180 mm was used. For the size of the nozzle, it has been reported that a 0.4 mm nozzle is frequently clogged with short fibers. On the other hand, 0.8 mm is too big to obtain precise fabrication. Manufacturer of Ruby nozzle, which can endure weariness from short fibers, recommended 0.6 mm. Therefore 0.6 mm was chosen in this study. The printing parameters employed were; nozzle extrusion temperature: 260 °C, heat bed temperature: 100 °C, deposition line (layer) height; 0.15 mm, and printing speed: 20 mm/sec. Carbon-fiber nylon filament (NylonX™ from Matterhackers, filament diameter: 2.85 mm) made of 4043D resin reinforced with chopped short carbon fibers (20% weight fraction) was used. The experiments were performed using SHIMAZU™ AGS-X HC universal testing machine (Figure 5b). A fully calibrated extensometer was utilized to precisely capture displacement change. All the specimens were loaded up to material failure at a displacement rate of 1 mm/min. The data acquisition rate was 10 Hz.

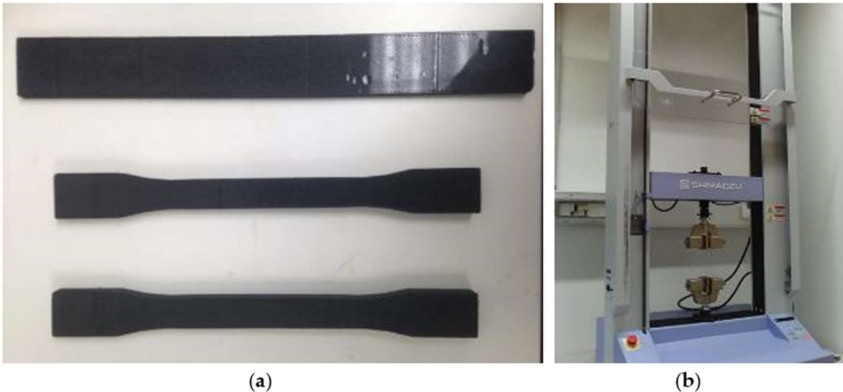

**Figure 5.** Test specimens (**a**) tensile ("dog bone", lower) and shear (rectangular, upper) and (**b**) SHIMAZU tensile machine.

Figure 6 shows the stress-strain curves for the specimens printed in the three different directions. Captured data from the machine was 10 points-average smoothed. As shown, the directional dependency of material properties is noted. Table 1 shows the mechanical properties of CFRP-Nylon (NylonX$^{TM}$) which were extracted from the stress-strain curves and used in the FE modeling analysis of FDM parts. Young's moduli and tensile yield strengths, as well as Poisson's ratios, were obtained for the printing directions of 0° and 90°. Moreover, shear strength was experimentally determined per ASTM D3518.

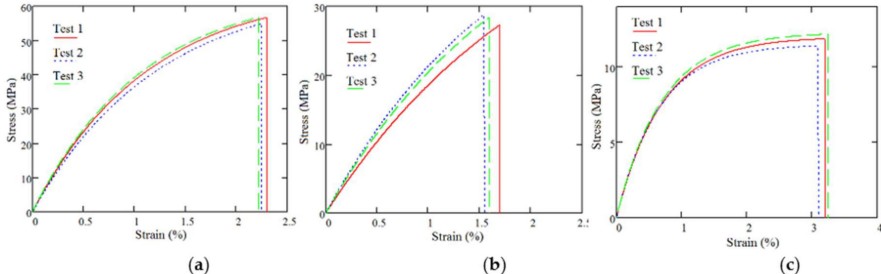

**Figure 6.** Stress vs. strain data for CFRP-Nylon printed at 0° (**a**), 90° (**b**), and ±45° (**c**).

**Table 1.** Mechanical properties of carbon fiber-reinforced polymer (CFRP)-Nylon.

| Property | Direction | CFRP-Nylon | ASTM |
|---|---|---|---|
| Young's Modulus | 0° | 4.14 GPa | D638 |
|  | 90° | 2.15 Gpa |  |
| Shear Modulus | ±45° | 1.12 Gpa | D3518 |
| Tensile Strength | 0° | 56.6 Mpa | D638 |
|  | 90° | 28.3 Mpa |  |
| Shear Strength | ±45° | 11.9 Mpa | D3518 |
| Elongation | 0° | 2.30% | D638 |
|  | 90° | 1.59% |  |
| Poisson's ratio | $\nu_{12}$ | 0.391 | D638 |
|  | $\nu_{21}$ | 0.203 |  |

### 3.2. Fiber Orientation

The major assumption underlying the proposed method is that short fibers are fully aligned with the direction of extrusion. Although the fiber orientation of parts was indirectly verified through lab tests, visual verification is required to understand the potential and limitation of the proposed method caused by the inherent drawback of short carbon fibers. The orientation of short fibers embedded in printed parts was investigated by an optical microscope. For this task, the specific method developed

by Bay and Tucker [35] was followed to characterize the short fiber orientation of the printed samples. Figure 7. Shows an SEM image of CFRP-Nylon fractured surface, as shown, most of the fibers are aligned in a single direction. Figure 8. shows an optical microscope image of a printed CFRP part. The methodology for fiber alignment analysis is based on the simple geometrical assumption that, ideally, cylindrical fiber should appear on the section as either circular or elliptical shape. However, as shown in Figure 9a, due to variations of fiber tip geometry, such as covering of the matrix material on the fiber tips surface or breakage of fiber tip surface, some of the cross-sectional shapes of fibers are not suitable for fiber alignment analysis—thus, those were excluded.

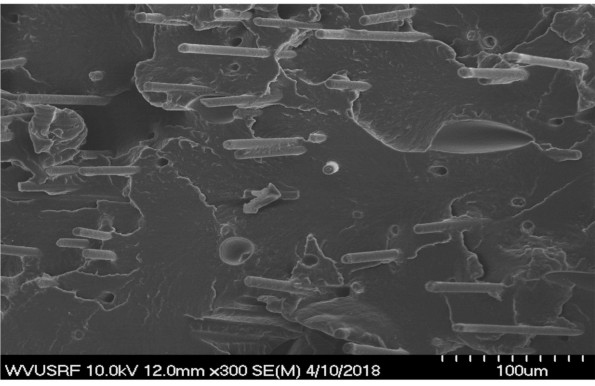

**Figure 7.** SEM image of CFRP specimen printed at 45° direction.

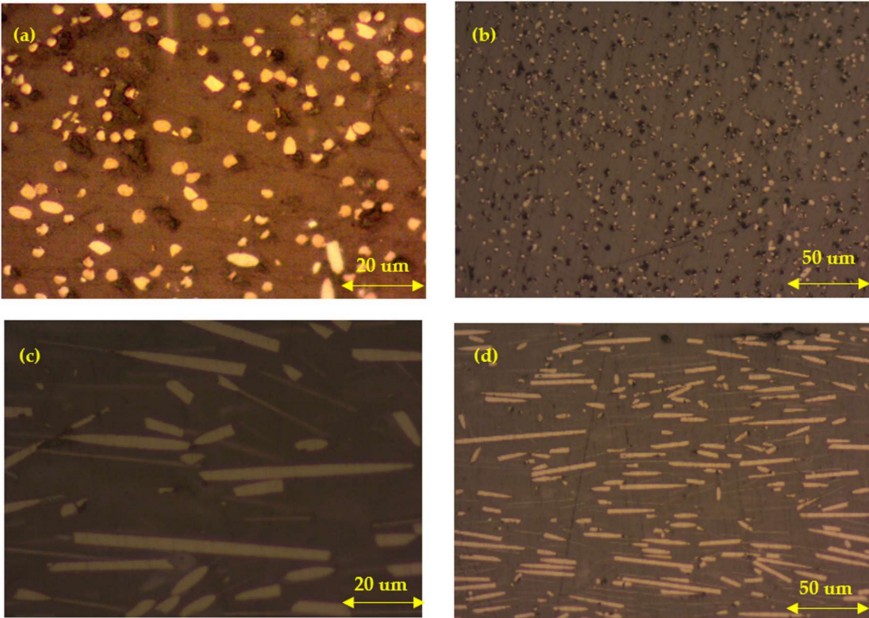

**Figure 8.** Sample micrographs of a polished specimen obtained from optical microscope: (**a**) Cross-section at 0° (printing direction), magnitude 50×; (**b**) cross-section at 0°, magnitude 20×; (**c**) cross-section at 90°, magnitude 50×; (**d**) cross-section at 90°, 20×.

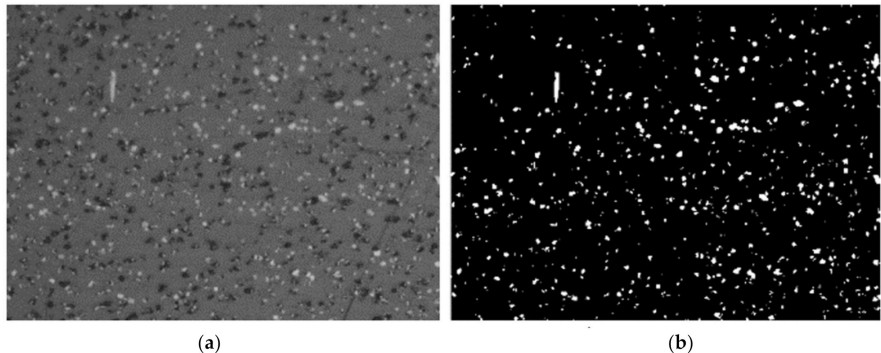

**Figure 9.** Pixel conversion to white/black. (**a**) original, (**b**) converted.

Assuming the length and diameter of the short carbon fibers are uniform as approximately 100 μm and 5 μm, the measurement was processed to find only the ends of the major axis. Each image was broken into pixels; each pixel has a value corresponding to the intensity of light at its Cartesian location. These digital images are first subjected to a threshold operation, making each pixel either black or white, as shown in Figure 9b.

The next step is to identify a group of pixels representing each fiber and determine the relevant dimensions. It measured the chord length in several directions and then took the maximum and minimum values as the major and minor axes, respectively. Using the cross-sectional area and major/minor diameters, each cross-section is checked to confirm they are roughly either circular or elliptical. If those fibers whose cross-sections are not circular or elliptical, they would not be used for fiber orientation analysis, and these fibers are filtered out, as shown in Figure 10b.

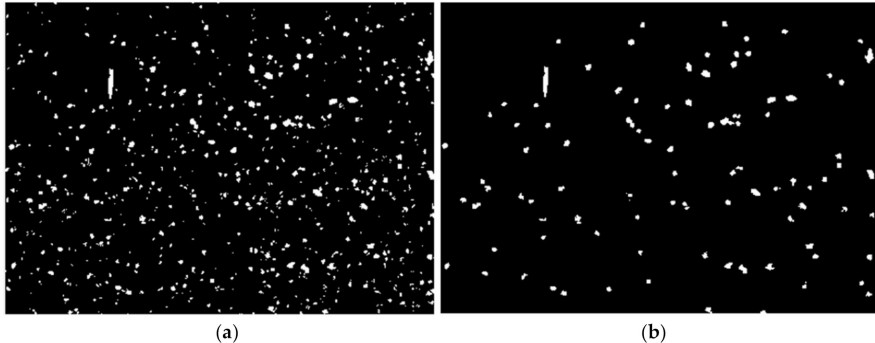

**Figure 10.** Filtering fibers with circular and elliptical cross-sections. (**a**) before, (**b**) after filtering.

Using the processed fiber images, such as Figure 10b, second-order tensors of fiber orientations in the directions of 1, 2, and 3, were computed. Table 2 shows the values of each component, as shown, the majority of the orientation tensor for the FFT sample is a11, an indication that practically most of the short carbon fibers are oriented in the 1-direction, which is the desirable load-bearing direction. This fiber orientation analysis indicated the inherent characteristic of high fiber orientation by the FDM process.

**Table 2.** Fiber orientation distribution of printed CFRP-Nylon sample.

| Orientation | a11 | a12 | a13 | a22 | a23 | a33 |
|---|---|---|---|---|---|---|
| % | 0.825 | 0.123 | 0.082 | 0.03 | 0.004 | 0.0056 |

## 4. Customized Tool-Path Development

FEA stress field is computed to obtain principal directions of elements, Figure 11 illustrates an example showing element-based principal directions of a cantilever beam under uniformly

distributed load. Centroids of elements are connected to create a printing path. The created printing path are aligned with the principal directions of individual elements. Square shell elements are utilized to reduce computational time by creating straight path lines. The size of the shell element is determined based on the printer extruder diameter. The small size of elements guaranteed a more precise printing path; however, if it is smaller than the extruder diameter, the extruding width is set to be invalid and linked to neighboring elements. Figure 12 shows a framework of the optimized tool-path development; more details are explained below.

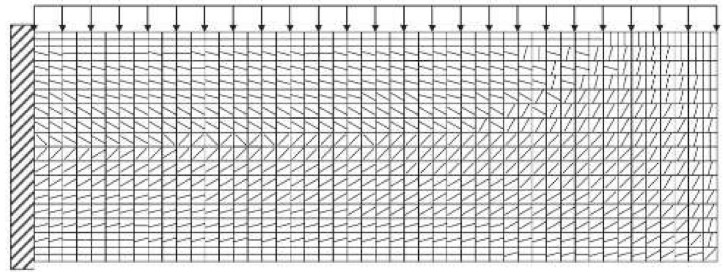

**Figure 11.** Principal directions of elements [36].

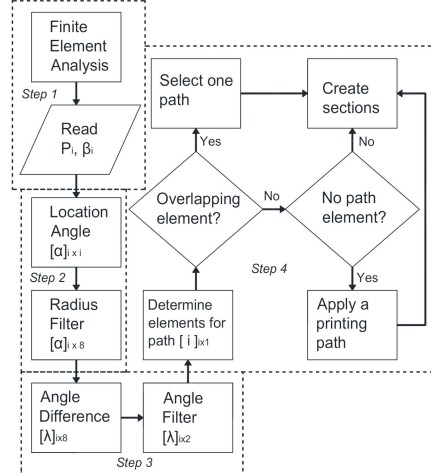

**Figure 12.** Optimized tool-path framework.

## 4.1. FE Analysis

In this step, principal stress $[P_i]$ and direction $[\beta_i]$ matrices of elements are generated from the FEA analysis output. Those are adopted as a basis to create a tool-path to assign fibers aligning with the principal direction of the structure under a given loading condition.

## 4.2. Location Angle and Radius Filter

The curvature of the printing path may affect the quality of the product. This means a sharp curve of the path usually produces more voids, which could lead to poor product quality. To address this issue, an effort to make the printing path straight has been made. Location angle matrix $[\alpha]_{ixi}$ is defined to set the geometrical relationship between two elements on the Cartesian coordinates system. As illustrated in Figure 13b, the angle between connecting lines of adjacent element centroids is computed. The center element (element 23) in Figure 13b is denoted as 'starting element' in this framework, as illustrated in Figure 13a. Then, a radius from the centroid of the starting element is defined to cover neighborhood elements only around the starting element. Neighborhood elements which are filtered by the radius are denoted as 'candidate elements'. Figure 13a describes the concept of elemental printing path connection.

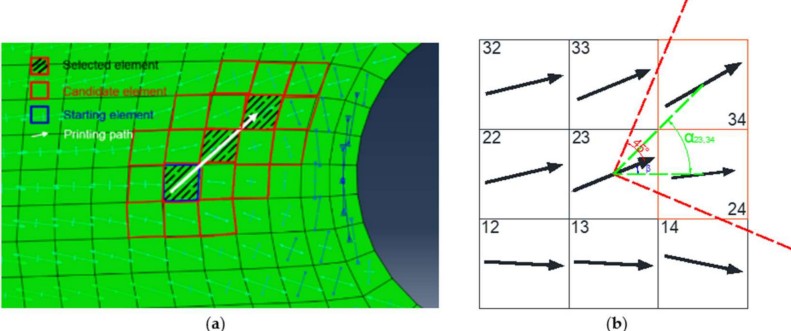

**Figure 13.** (**a**) Concept of elemental printing path connection; (**b**) concept of angle difference filtering.

### 4.3. Angle Difference Filter

Since the printing path must align with the principal directions of elements, the next connecting elements are selected by the principal direction of the starting element. In this step, the angle difference filter is defined as follows.

$$[\lambda]_{ix8} \rightarrow [\lambda]_{ix2} \text{ if } [\lambda]_{ix8} < 45° \text{ Where, } [\lambda]_{ix8} = [\alpha]_{ix8} - [\beta]_i$$

To keep only two elements as candidate elements, ±45° is added to the principal direction of the starting element. As shown in Figure 13b, the red boundary line indicates the final two elements for the printing path connection.

### 4.4. Creation of Printing Path and Sections

From the selected two candidate elements (e.g., Figure 13b), the element with larger principal stress is finally selected to be connected. If there are no candidate elements whose angle differences are less than 45 degrees, the printing path stops the connection. Through the procedure, a set of connecting lines is then created and referred to as a printing path section. Next, the contour-parallel or direct-parallel printing method is determined based on the computed path line and curvature of the path. If a path line moves to align with the product profile, the contour-parallel method is applied; otherwise, the direct-parallel method is applied. For the direct-parallel method, the angle of the infill pattern is determined by the average principal direction of elements occupied the section. To avoid overlapping of printing paths, if an element is selected multiple times for the next connecting element, the principal stresses of starting elements on each path are checked. Then, a printing path with larger principal stress is chosen to continue. Elements covered by similar patterns create a printing section. Printing path of elements having the end of the path, thus, creates section boundaries. Figure 14 shows the workflow of section generation and corresponding printing path.

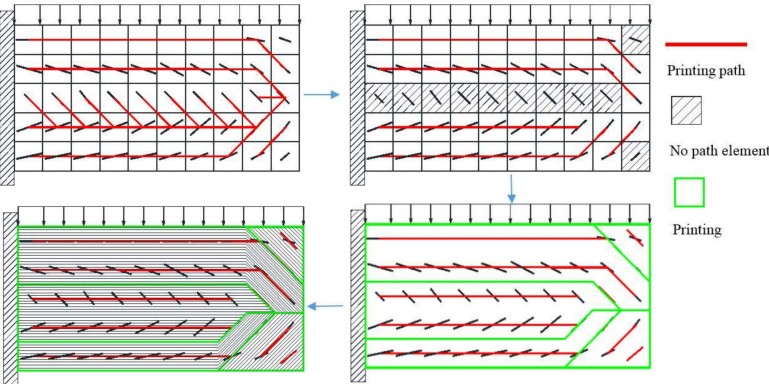

**Figure 14.** Principle of printing path and section development.

In contrast, if an element is not chosen by any path, groups of those elements are merged into neighborhood sections. For those elements, the stress level is usually low, and thus, any printing pattern is allowed, since it has a low impact on the strength of the final product. However, to achieve better product quality a uniform single printing path is applied to the merging section rather than generating printing section boundaries, which could lead to producing voids during the extruding. Lastly, 5% of overlapping is applied to guarantee adequate material bonding. Figure 15 shows an example of an optimized tool-path for a plate with a circular hole under tensile loading.

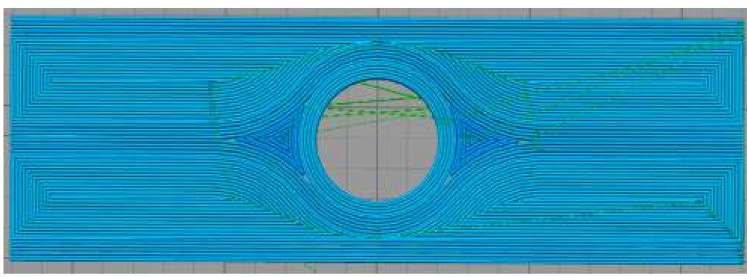

**Figure 15.** Optimized tool-path.

### 4.5. FE Modeling of Printed Object

Generally, slicer programs, such as Simplify3D, generate G-codes for FDM printing, including controlling of extruder movement. In this research, G-codes are then converted into ABAQUS$^{TM}$ input to define outer boundary lines, material orientations of elements, and mesh size. Specific Matlab$^{TM}$ codes were created to process this step, which provides angle values in the Cartesian coordinates of the FE elements. Material orientations were directly computed from G-codes, as illustrated in Figure 16. Moreover, the orthotropic material properties matrix was applied to elements from the experimental data in Section 3.1.

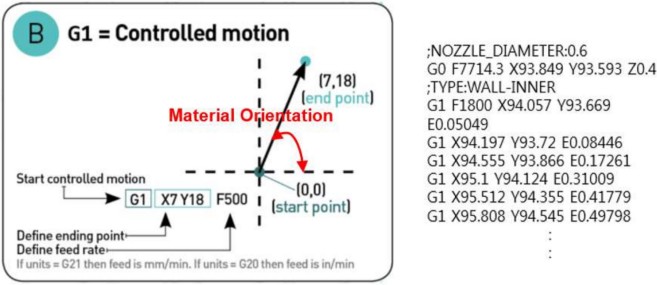

**Figure 16.** Determination of material orientation.

## 5. Case Study-Hole Plate

### 5.1. Problem Statement—Hole Plate

Stress concentration in a plate with a hole was chosen to demonstrate the advantage of the customized tool-path printing method. Figure 17 shows the maximum principal stress flow and a printed CFRP sample.

### 5.2. Tool Path Development—Hole Plate

To compute principal directions of each element, FEA stress analyses were carried out using ABAQUS$^{TM}$. For simplicity, 2D shell elements were applied. As described earlier, square elements were employed to connect elements to generate a custom tool-path for printing paths and sections. The height, width, and radius of the hole are 120 mm, 40 mm, and 10 mm, respectively.

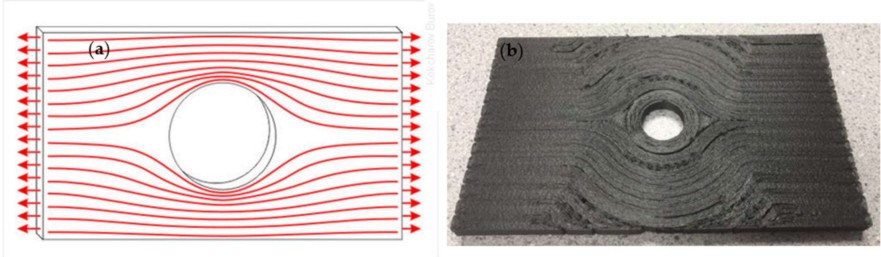

**Figure 17.** (**a**) Stress flow and (**b**) an FDM built CFRP sample of a plate with a circular hole.

Figure 18 shows the maximum principal stress field and the corresponding principal directions of individual elements. As expected, high stresses occurred around the center hole, and their principal directions are aligned with the hole (shown as short length arrows in Figure 18b).

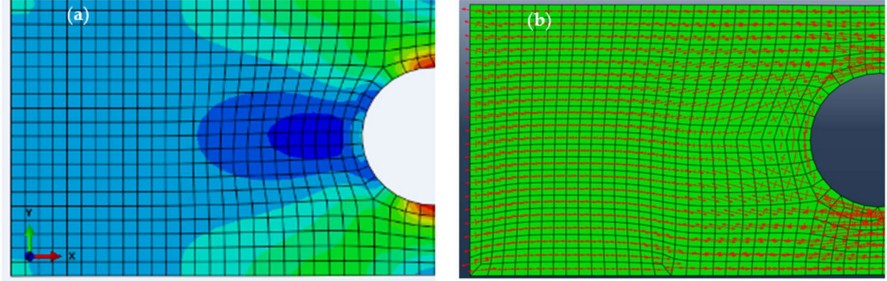

**Figure 18.** (**a**) maximum principal stress field and (**b**) principal directions of the plate with the circular hole under tensile loading.

Figure 19 shows the section division of the plate for tool-path based on the proposed method. Each element is connected to create sections. A customized tool-path is created for each section based on principal directions of elements. For example, the tool-path around the center hole is aligned with the circle as principal directions are parallel to the circle. Moreover, the low-stress region in Figure 19a has a 0° uniform tool-path, which is the same as the tool-path in other major regions. Figure 19b shows the complete tool-path development of the whole plate.

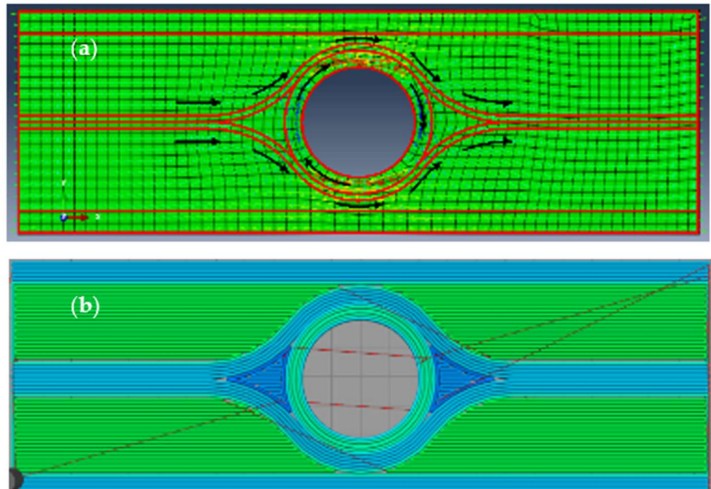

**Figure 19.** (**a**) Section division for tool-path and (**b**) complete optimized tool-path of stress concentration plate.

### 5.3. FE Modeling—Hole Plate

Figure 20 shows the maximum principal stress distribution of the plate with the updated orthotropic material properties induced by the customized tool-path. Non-linear explicit analysis with eight brick elements was performed using ABAQUS$^{TM}$. The thickness of the model was reduced by applying only two layers to reduce the running time. As shown, high stresses (with 8% less magnitude) still occurred around the hole, but were much confined compared to the stress distribution of plate with isotropic material (Figure 18a). Moreover, as an illustration, to show the benefit of tool-path optimization assisted by FEA, Figure 21a is a uniform 0° printing path (i.e., no optimization), and Figure 21b is the optimized printing path. Figure 21c,d show its corresponding shear strain distributions. As shown, for the case of no optimization (Figure 21c), high shear strains at the edge of the hole will be the likely failure initiation site. As for the optimized case (Figure 21d), the magnitude of the highest shear strain is 16% less than the non-optimized case; moreover, the high shear strain regions are away from the hole edge, due to the optimized printing paths.

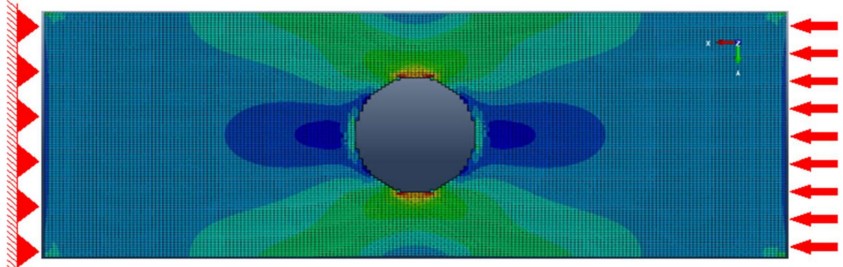

**Figure 20.** Stress distribution of the plate printed based on optimized tool-path.

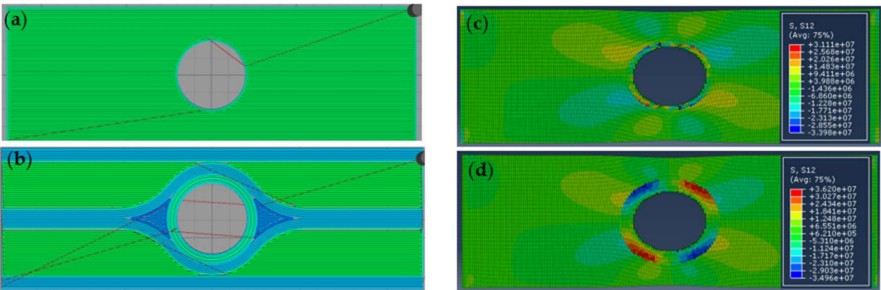

**Figure 21.** (**a**) 0° uniform; (**b**) customized tool-path; (**c**) shear strain distribution of samples printed by 0° uniform; (**d**) customized tool-path.

### 5.4. Tensile Test—Hole Plate

As shown in Figure 22, relevant tensile tests for the printed plates were conducted [37] to validate FEA simulation results. The loading rate was 1 mm/min, with data recorded at every 0.01 sec. Using the force and displacement data, the stiffness response was computed. Figure 22b shows load vs. displacement curves for each case. Averaged test results of each case are shown in Table 3.

The plate printed by the optimized tool-path shows the highest stiffness (6% higher than the direct-parallel (0°) case). However, there is no significant improvement in overall failure strength. Figure 23 represents the comparison of stiffness response results between FEA simulations and tensile tests. Discrepancies between analysis and test for 0°, contour, ±45°, and optimized tool-path show 9.4%, 10.1%, 10.2%, and 11%, respectively, which suggests the optimized tool-path generated slightly more voids than other conventional tool-path methods.

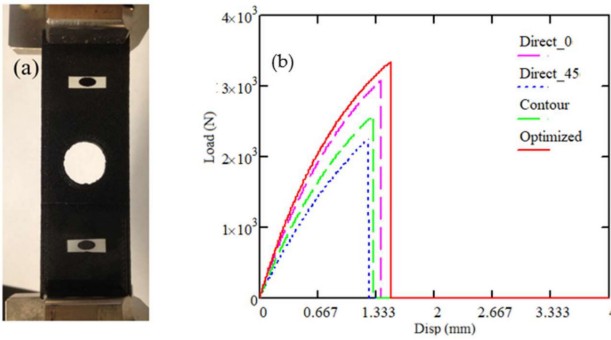

**Figure 22.** (**a**) Tensile test and (**b**) test result of stress concentration plate.

**Table 3.** Tensile test results of stress concentration plate printed by various tool-paths.

|  | Stiffness Response (N/mm) | Failure Strength (N) | Max. Displacement (mm) |
|---|---|---|---|
| Direct-parallel (0°) | 2544.0 | 3087 | 1.39 |
| Contour-parallel | 2161.7 | 2592 | 1.30 |
| Direct-parallel (±45°) | 1983.4 | 2262 | 1.25 |
| Optimized | 2687.0 | 3349 | 1.50 |

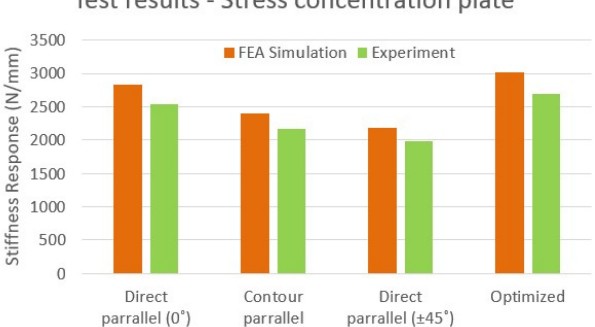

**Figure 23.** Comparison of stiffness response between finite element analyses (FEA) simulations and experiments.

## 6. Case Study—Spur Gear

### 6.1. Problem Statement—Spur Gear

Gears are mechanical components used for transmitting motion and torque from one shaft to another. A spur gear is the most common type of gear, and the tooth regions are prone to failure, due to high contacting stresses. Figure 24 shows an example of a failed Nylon spur gear in a ball-milling machine.

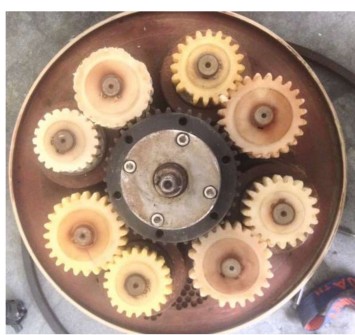

**Figure 24.** Spur gear failure in a ball-milling machine.

### 6.2. Determination of Fiber Orientations—Spur Gear

Using ABAQUS$^{TM}$, FEA analyses were carried out for the pre-described spur gear geometry under the prescribed loading and boundary conditions. The central main steel gear (Figure 24) was considered as rigid in the FE analysis (shown as the white section in Figure 25a). Figure 25 shows the maximum principal stress field (Figure 25a) and principal directions (Figure 25b) of elements at the spur gear tooth region.

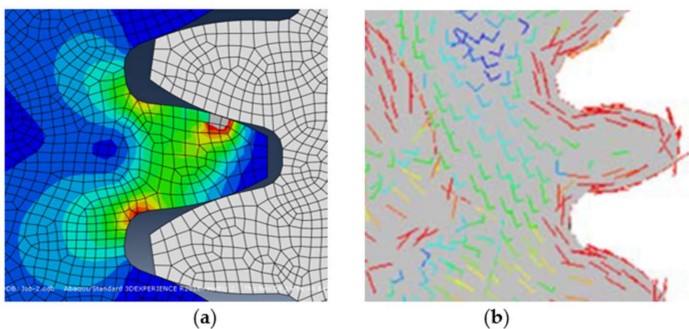

**Figure 25.** (**a**) Maximum principal stresses; (**b**) principle directions at spur gear tooth region.

### 6.3. Tool Path Development—Spur Gear

Figure 26 shows principal directions and section divisions for FDM printing. Different printing patterns were applied to individual sections based on output from the steps described in Section 4. Figure 27 shows the optimized tool-path for the gear. Noted that principal directions of elements in the tooth region are oriented along with the gear tooth profile such that the ideal way to reinforce the gear is to align fibers with the tooth profile in the high-stress region.

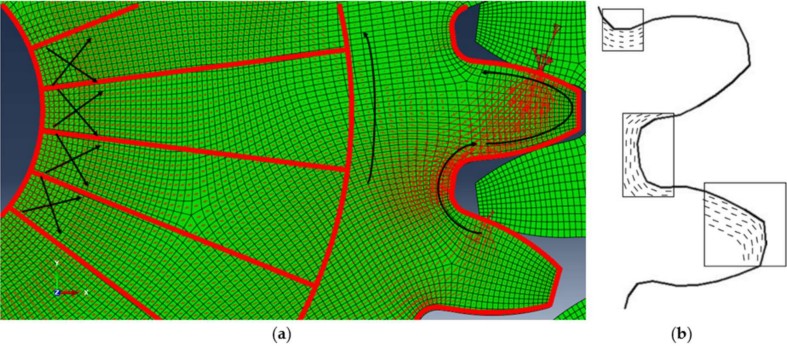

**Figure 26.** (**a**) Section division of spur gear for tool-path; (**b**) principal directions of teeth section.

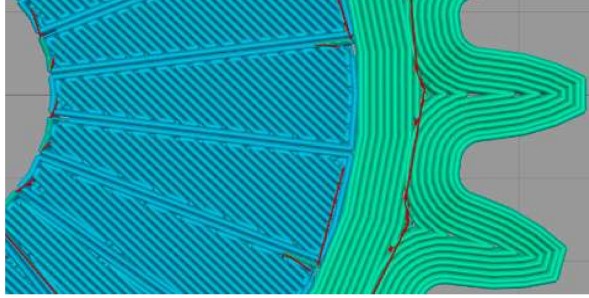

**Figure 27.** Optimized tool-path for spur gear.

### 6.4. FE Modeling—Spur Gear

Coordinate information was exported from G-codes and converted into material orientations of individual elements. Only the initial eight layers were applied to reduce the running time. A total of 180,000 Hexahedron elements having an extruder width (0.6 mm) was generated. Figure 28a is a captured image from the process of mesh generation. Figure 28b shows a FEA simulation of the experiment test (Figure 29). The pushing hammer was treated as rigid, vertical displacement of 1.0 mm was applied.

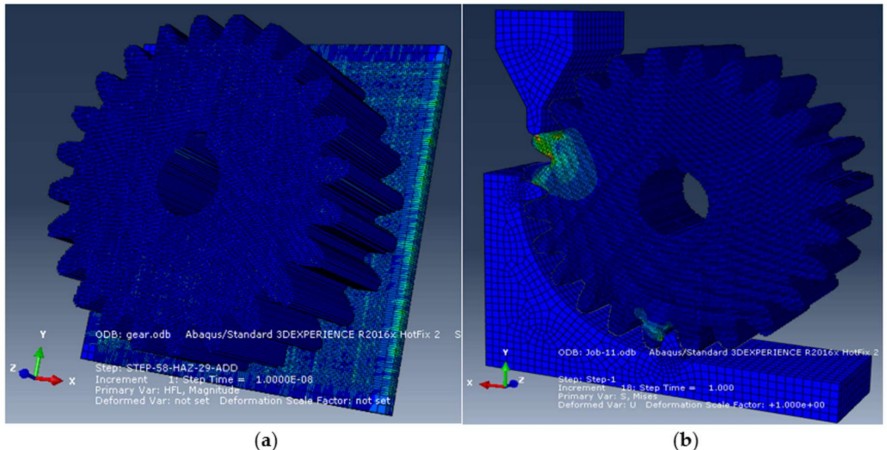

**Figure 28.** (**a**) FE modeling of spur gear; (**b**) FE simulation of the experimental test in Figure 29.

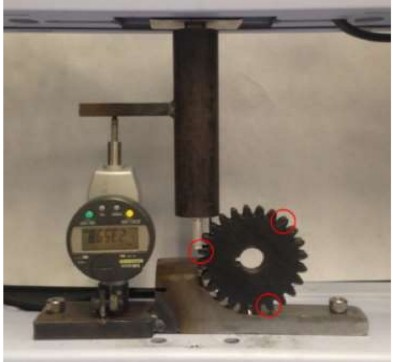

**Figure 29.** Spur gear stiffness test for CFRP-Nylon.

Stiffness responses from several different build orientations, including optimal fiber orientations, unidirectional orientation with [0°], [±45°]s, and contour direction were obtained to compare structural performance.

### 6.5. Stiffness Test—Spur Gear

To validate the FEA simulation results, relevant compression tests for spur gear were conducted, as shown in Figure 29. As previously discussed, high stress was generated in the teeth region. The corresponding displacement was measured by a digital dial indicator with a resolution of 0.001 mm. The loading rate is 1 mm/min, and the data was recorded every 10 s.

Figure 30 shows load vs. displacement curves for each case. Averaged test results at three different locations are shown in Table 4. For repeatability of the test results, three different locations were tested, and variation was less than 1%.

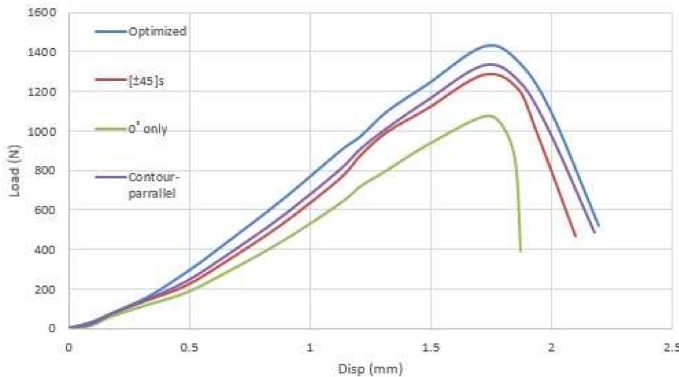

**Figure 30.** Load vs. displacement curves of CFRP spur printed by various tool-path.

**Table 4.** Test results of stiffness response of CFRP-Nylon.

|  | Stiffness Response (N/mm) | Failure Strength (N) | Max Disp (mm) |
| --- | --- | --- | --- |
| Contour | 778.6 | 1330 | 2.18 |
| Optimized | 834.6 | 1430 | 2.19 |
| [±45°]s | 754.6 | 1287 | 2.10 |
| 0° only | 607.0 | 1073 | 1.87 |

As shown in Figure 30, gear printed by the optimized tool-path shows the highest stiffness response and failure strength. It shows approximately 7% higher in stiffness response than gear printed by contour-parallel. For the failure strength, optimized gear shows 8% higher than contour-parallel.

For the result of FEA simulations, the principal stresses at the critical location of each case were measured, and the stiffness response was computed. The gear with the optimal fiber orientation shows the highest stiffness response of 981 N/mm. For other cases, contour-parallel, [±45°]s, and [0°] showed 894 N/mm, 847 N/mm, and 682 N/mm, respectively. Approximately 9% improvement of stiffness was observed when compared with the contour-parallel tool-path. Figure 31 shows a comparison of stiffness response results between FEA simulation and laboratory experiments. The discrepancy between FEA modeling analyses and experiments is 12% for the optimized tool-path and 9% for other conventional tool-path.

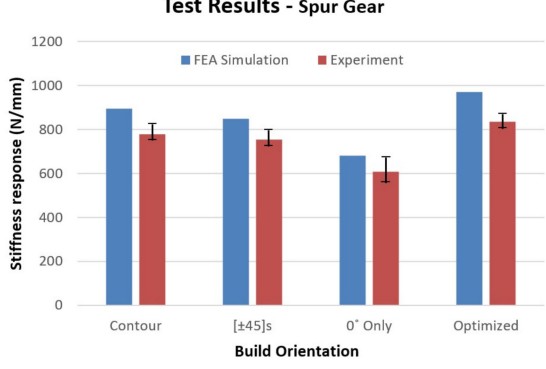

**Figure 31.** Comparison of results between FEA simulation and experimental tests.

## 7. Summary

A 3D printing methodology for fused deposition modeling (FDM) integrating fiber placement and tool-path development has been proposed. It starts with FEA to obtain principal stress fields and principal directions. Optimal fiber orientations of individual elements were determined. Using the stress output data from ABAQUS$^{TM}$, a tool-path optimization algorithm to maximize the effect

of fiber reinforcement of CFRP was developed. To predict the mechanical behavior of the printed parts, G-codes from the tool-path were used to model printed objects.

The proposed methodology demonstrates that the optimized tool-path can be applied to the 3D printer to extrude fibers aligned with principal directions. The flow distribution of printed fibers was verified by image analysis, which showed that approximately 83% of fibers were oriented as intended. Both FEA and preliminary experimental case study results show that CFRP-Nylon parts printed by the optimized tool-path achieved approximately 8% improvement in structural performance over parts printed at regular uniform printing direction. Associated experimental test results represent 15% lower stiffness responses than those from FEA predictions. Although 8% improvement is insignificant, nevertheless, with further AM printing process optimization to better control the fiber orientation, as well as advanced material development, the proposed customized tool-path method presented in this paper can be utilized as a design and printing methodology for CFRP structural parts by FDM technology.

The proposed methodology can be extended to other materials. Moreover, with the development of a dual extruder system, dual material optimization would be another topic to overcome the brittle nature of CFRP. To predict their durability more accurately, fatigue tests are required. Due to the limitation of the current FDM printing system, only in-plane tool-path optimization has been allowed in this research. However, if AM with a tilted bed is fully developed, it may provide research opportunities for three-dimensional tool-path optimization.

**Author Contributions:** Conceptualization, B.S.K.; methodology, J.K.; software, J.K.; validation, J.K.; formal analysis, J.K.; investigation, J.K.; resources, J.K.; data curation, J.K.; writing—original draft preparation, J.K.; writing—review and editing, B.S.K. and J.K.; visualization, J.K.; supervision, B.S.K.; project administration, B.S.K.; funding acquisition, B.S.K. All authors have read and agreed to the published version of the manuscript.

**Funding:** This research received no external funding.

**Conflicts of Interest:** The authors declare no conflict of interest.

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
