# Peer review of "Enhancing Structural Performance of Short Fiber Reinforced Objects through Customized Tool-Path"

_applsci, doi:10.3390/app10228168_

Round 1
Reviewer 1 Report
The study presents a methodology to evaluate and take into account during the mechanical analysis, the anisotropic behavior of parts manufactured by FDM employing CFRP. The authors establish a procedure and use experimental and numerical procedures to evaluate the method. The paper needs a deep modification to ensure academic rigor and some rewriting to reduce the similarity with other works (Turnitin coincidence:19%, the reviewer suggested less than 15%).
The English must be improved in the entire document as an example only in the abstract, you can find the following errors:
Line 17: "anisotropic" instead of "aniostopy".
Line 18: "the FDM" instead of "FDM"
Line 22: "the mechanical" instead of "mechanical"
Line 22: "the AM" instead of "AM"
Line 22: "the 3D" instead of "3D"
Line 28: "the methodology" instead of "methodology"
regarding the document:
- The authors analyze the fiber orientation based on a nozzle diameter of 0.6mm. The fiber orientation during the injection process varies along with the thickness. The reviewer suggests a discussión of the effect of the nozzle diameter in the fiber direction or sensitivity analysis of the fiber orientation with different nozzle diameter (0.4,0.6,0.8,1mm)
- The numerical model is not defined, Implicit or explicit analysis, linear or non-linear, type of element, and the number of elements, element length, plane-stress or plane strain, etc...
- The short carbon fiber is defined by a linear stress-strain response, but thermoplastic uses a nonlinear response. How are defined the interaction between these materials in the same unit cell? Is defined the thermoplastic response as Isotropic or kinematic hardening?.
- Figures 18-21, the legend with stress tensor does not appear, please introduce the legend to compare results between the proposed model and 0º uniform material analysis.
- Line 365, is Figure 22 instead of 25
- In table 23 introduce the value of normalized error between FEA(proposed model)/FEA(0ª uniform)/Experimental, to compare between then.
- Figure 31, introduce the value of normalized error.
- In Summary, the authors concluded: "The proposed methodology demonstrates that the optimized tool-path extrudes fibers aligned with
474 principal directions with high accuracy". This accuracy must be measured by normalized error.
Author Response
The authors analyze the fiber orientation based on a nozzle diameter of 0.6mm. The fiber orientation during the injection process varies along with the thickness. The reviewer suggests a discussión of the effect of the nozzle diameter in the fiber direction or sensitivity analysis of the fiber orientation with different nozzle diameter (0.4,0.6,0.8,1mm)
-> Only 0.6mm of ruby nozzle worked for CFRP at the time of study. Please see line 150
- The numerical model is not defined, Implicit or explicit analysis, linear or non-linear, type of element, and the number of elements, element length, plane-stress or plane strain, etc...
-> See line 345
- The short carbon fiber is defined by a linear stress-strain response, but thermoplastic uses a nonlinear response. How are defined the interaction between these materials in the same unit cell? Is defined the thermoplastic response as Isotropic or kinematic hardening?.
-> We used a commercial filament which provides mechanical properties, it was verified by our lab test again to investigate orthotropic properties. Then experimental data which shows non-linear (but brittle) were discretized and utilized as input for the FEA modeling. in this case, interaction is not needed to be considered. Full bonding was assumed in FEA, but orthotropic properties from experiment was applied to account for the weakness of strength between layers.
- Figures 18-21, the legend with stress tensor does not appear, please introduce the legend to compare results between the proposed model and 0º uniform material analysis.
->figure 18-19 is for principal direction, figure 20 shows concentrated stress region. legend was attached to figure 21
- Line 365, is Figure 22 instead of 25
->fixed
- In table 23 introduce the value of normalized error between FEA(proposed model)/FEA(0ª uniform)/Experimental, to compare between then.
->added
- Figure 31, introduce the value of normalized error.
- In Summary, the authors concluded: "The proposed methodology demonstrates that the optimized tool-path extrudes fibers aligned with
474 principal directions with high accuracy". This accuracy must be measured by normalized error.
->added and fixed
Reviewer 2 Report
This paper fills a gap within the FDM technical field.
The combination of experimental and FEA approach is very wise and provide the reader a good methodology for design optimisation for fibre CFRP 3D printing.
line 117: in Figure 3 case (b) is missing in figure caption
line 136: ASTM D638 Type I has been chosen for the tensile strength measurment. It is not clear from the photo of the sample if contour were used during the printing strategy. Please add in the manuscript if a specific slicing sofware was used (i.e. Simplify 3D) and if countour were used and how much?
It is clear that for 90 orientation if contour are present and oriented in the 0 direction (along the tensile testing direction) this might have an influence.
Same question for the test speciment for shear measurment using 45/45 printing strategy: how many contour were used?
Please update manuscript accordingly.
line 165: Table 1. As general comment for tensile testing, did you use an extensometer during testing for the calculation of the Moduli? If not how reliable are the modulus calculated and elongation mentionned from the Stress/Strain curve presented?
lin 355: Figure 21 caption a,b, c, d are not visible on the figure
General comment: A 0.6 mm nozzle has been chosen for the printing. Looking at the optimised path there are potential location where the 3D printer need to print very short path and/or fill small space, this might create voids. Comment on the prorosity of the produce parts need to be added in the manuscript. Does optimised path increase the overall porosity and/or pore size/void compare to a "classical" path. Can this be mimimised with smaller nozzle diameter ?
Author Response
It is clear that for 90 orientation if contour are present and oriented in the 0 direction (along the tensile testing direction) this might have an influence.
Same question for the test speciment for shear measurment using 45/45 printing strategy: how many contour were used?
-> We were aware of it. The visible line from the picture is a mechanical stain from CNC cutting saw. it is not printing contour. See line 137.
Please update manuscript accordingly.
line 165: Table 1. As general comment for tensile testing, did you use an extensometer during testing for the calculation of the Moduli? If not how reliable are the modulus calculated and elongation mentionned from the Stress/Strain curve presented?
-> extensometer was fully calibrated with ASTM standard aluminum and steel specimens.
lin 355: Figure 21 caption a,b, c, d are not visible on the figure
-> fixed
General comment: A 0.6 mm nozzle has been chosen for the printing. Looking at the optimised path there are potential location where the 3D printer need to print very short path and/or fill small space, this might create voids. Comment on the prorosity of the produce parts need to be added in the manuscript. Does optimised path increase the overall porosity and/or pore size/void compare to a "classical" path. Can this be mimimised with smaller nozzle diameter ?
-> only 0.6mm of ruby nozzle was avalaiblae for CFRP at the time of study. see line 152. We controlled curvature of printing path to reduce voids. see line 240
Round 2
Reviewer 1 Report
The paper deserves to be published